# THINKING INTO THE FUTURE: LATENT LOOKAHEAD TRAINING FOR LANGUAGE MODELS

## ABSTRACT

Autoregressive language models trained with next-token prediction generate text by sampling one discrete token at a time. Although very scalable, this objective forces the model at every step to commit to a singular choice, preventing it from exploring or reflecting upon multiple plausible continuations. Furthermore, the compute allocation across tokens is uniform; every token is formed based on a single forward-pass, potentially limiting the model's expressiveness in cases where difficult tokens require inherently more compute. Towards addressing these limitations, we introduce *latent lookahead*, a training strategy that enables models to "think" before generating: at selected positions in the sequence, before committing to the next token, the model performs a multi-step lookahead in latent space. More precisely, instead of sampling future tokens, we leverage the network's latent space by recursively feeding its hidden states back into the context for $\tau$ steps, investing more compute on predicting that token. This produces $\tau$ latent predictions that are supervised against the next $\tau$ ground-truth tokens, encouraging the model to "lookahead" and refine its prediction. We show that latent lookahead substantially outperforms autoregressive baselines on planning tasks such as maze solving, Sudoku, and ProsQA, where foresight is essential. Finally, we demonstrate how to endow pretrained models with this ability during supervised fine-tuning and evaluate the resulting models on standard reasoning benchmarks.

## 1 INTRODUCTION

At the core of modern Large Language Models (LLMs) lies an autoregressive training paradigm based on two core ingredients: next-token prediction (NTP) Shannon (1948) and teacher-forcing. In NTP, the model maximizes the log-likelihood of the next token $x_t$, learning to predict each token given the preceding ground-truth context. Teacher-forcing supplies that ground-truth prefix $x_{<t}$ at every step during training rather than the model's own sampled outputs. During inference, the model samples from its hidden states the next token based on the context that it has generated so far. This paradigm is simple, scalable, and largely self-supervised, but it introduces two problems. First, it provides only a one-step training signal, thus biasing the model towards a narrow future horizon. As a result, models may behave myopically, predicting the next token without explicitly reasoning about the downstream consequences several steps ahead. Second, the model is forced to decode (through sampling) the token from its hidden states at every generation step. While this is flawless when the model is certain about its predictions, it may introduce errors when faced with uncertainty. Instead, it would be ideal to consider all the possible viable options, rather than being forced to commit immediately. These two problems compound together, where the next token prediction training together with forced sampling may introduce sources of errors, such as reaching dead-ends and wrong chain-of-thought (Dziri et al., 2023; Bachmann & Nagarajan, 2024).

However, many tasks in reasoning and decision-making require planning: evaluating alternative futures, and selecting actions that pay off over multiple steps. Humans routinely imagine possible continuations before committing to the next move. As we enter the era of agentic AI, where systems must execute structured sequences of actions, LLMs likewise need the ability to speculate about the consequences of their actions by *looking ahead* rather than reacting one token at a time. As a motivating example, consider the $4 \times 4$ Sudoku in Figure 2. In this simplified version of the game, the player needs to fill the blanks " _ " such that every row, column, and $2 \times 2$ grid contains the numbers $1, 2, 3, 4$. In the example, the first blank cell admits two candidates $\{1, 3\}$ under local

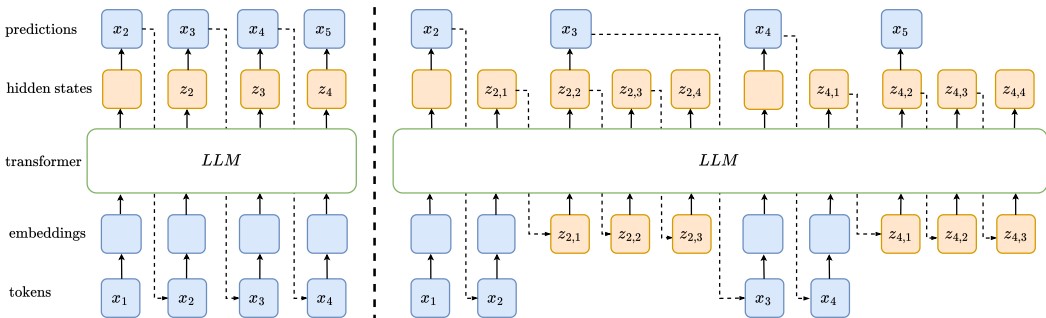

Figure 1: **Standard autoregressive inference vs latent lookahead.** Left: in standard next token prediction, the model samples from the hidden state of the latest generated token after applying the final unembedding head, and appends the generated token to the context. Right: in our approach, the model enters the latent lookahead thinking, where the hidden states are fed directly into the context instead of sampled visible tokens. This procedure is repeated $\tau$ times, and then the visible token is sampled from the first latent position. In the figure above, the tokens $x_2$ and $x_4$ are selected, $\tau = 3$, and $z_{i,j}$ indicates the $j$-th latent token relative to the $i$-th visible token. See also Fig. 3

row/column constraints; committing immediately is unjustified, and more computation is useful to accurately predict it. Using the extra compute to look a few steps ahead propagates the implications of each choice and quickly disambiguates which branch leads to a consistent continuation. In the portrayed example, it is easy to see that in the third cell 3 is the only viable option. This realization retrospectively constrains the original first cell to the single option 1. This also illustrates why compute should be allocated non-uniformly; predicting the first cell is inherently more difficult.

Motivated by this, we propose *latent lookahead*, a framework that allows for more flexible compute allocation per token, mitigating the described sources of errors. The model is illustrated in Fig. 1. Before emitting the next token, the model unrolls its hidden state $\tau$ steps into the future, by iteratively feeding predicted latent states back into the context. This avoids sampling, allowing more computation for the model before emitting the token. After $\tau$ steps, the model predicts the next token from the hidden states of the first latent, which can be iteratively refined as we allow for bi-directional attention between latent tokens, as illustrated in Fig. 3. To enforce the lookahead, the $\tau$ latent predictions are supervised against the next $\tau$ ground-truth tokens. This training procedure explicitly trains the model for lookahead capability.

Our work is inspired by the observation that increasing the computation that is performed by the model is beneficial during training and inference (known as test-time scaling): this can be done in various ways. Crucial for our model, either in the form of Chain-of-Thought (CoT), or with extra dummy tokens appended to the context, or latently, by looping multiple times over the transformer's layers. We review these methods in Sec. 2. A key design principle of our approach is interpretability: compared to existing proposed models in the realm of latent computation Hao et al. (2024); Deng et al. (2024), each latent token is explicitly supervised, and the model produces explicit CoTs, ensuring that the latent computation remains transparent. This is crucial, since fully latent models risk worsening the difficulty of interpreting LLM chain-of-thoughts, which are already known to be frequently unfaithful (Lanham et al., 2023; Chen et al., 2025). Concretely, we make the following contributions:

- We introduce the latent-lookahead framework and detail its training and inference procedure. See Figures 1 and 3. Latent Lookahead supports latent thinking at multiple steps (denoted by $n$). We design an attention mask that enables parallel generation of latent thoughts while preserving consistency between training and inference, reducing the cost to $\mathcal{O}(\tau)$ forward passes from the naive $\mathcal{O}(n\tau)$ of sequential generation.

- We demonstrate that latent lookahead is particularly effective on planning-oriented tasks, achieving substantial gains across three tasks: $4 \times 4$ and $9 \times 9$ Sudoku, ProsQA, and Maze. Notably, our method pushes $9 \times 9$ Sudoku accuracy from 15% to 35% over standard autoregressive baselines.

Figure 2: Lookahead behaviour when solving a Sudoku. In the first slot, both 1 and 3 are viable options. However, when thinking ahead to the second empty slot, where 3 is the only plausible entry, it is easy to realize that 1 is the right choice for the first slot.

- We evaluate the applicability of our framework by equipping existing base language models with latent lookahead during supervised fine-tuning (SFT). On math and logical reasoning benchmarks, latent lookahead obtains mixed results, overall matching autoregressive baselines. We report results with Olmo 2 (1B) and Qwen 2.5 (0.5B).
- We provide interpretation of the latent tokens that are generated, showing how they encapsulate the notion of *superpositions* of states, confirming results from previous theoretical works on latent CoT (Zhu et al., 2025).

## 2   RELATED WORK

Our work is inspired by and lies at the intersection of several foundational works in the fields of latent reasoning, multi-token prediction, and expanding the model's computation.

**Latent Computation.** Several works propose reusing hidden states to enhance reasoning. Hao et al. (2024) incorporate CoT traces into continuous "thinking tokens", autoregressively generated and appended to the context. The key idea is that the model can "think" in latent space without producing visible tokens, thus exploring multiple options. Zhu et al. (2025) formalize learning by superposition, showing gains in tasks like graph reachability. Another direction employs looped transformers (Giannou et al., 2023; Fan et al., 2024), which recursively update hidden states *à la* recurrent networks. These show promising reasoning results (Saunshi et al., 2025) and enable test-time scaling via arbitrary recursive steps(Geiping et al., 2025), but at the cost of FLOPs equivalent to much deeper models. In contrast, our approach applies recursion only at selected positions.

**Multi-Token Prediction (MTP).** The fact that the latent lookahead supervises the latent tokens with future tokens is reminiscent of multi-token prediction (MTP) (Gloeckle et al., 2024). In this line of research, the closest work to ours is Deepseek's MTP (Liu et al., 2024), where the model's final hidden states are re-used sequentially in the next steps. However, auxiliary modules for each next token are used, and their approach is fully causal. In latent lookahead the aim is to invest the extra compute to predict the next token also during inference. Also, our model is non-causal, so the model can refine the prediction of the next token while "thinking" about the future tokens. Other works have looked into MTP for faster inference (Stern et al., 2018; Cai et al., 2024), and in particular there is a rich line of work using speculative decoding (Li et al., 2024; Samragh et al., 2025).

**Expanding the context with extra tokens.** Wang et al. (2023) uses "planning tokens" to augment the CoT traces, prepending a number of tokens before each reasoning step. The planning tokens are generated with different variants from the hidden states of a base LLM. Goyal et al. (2023), Herel & Mikolov (2024) (for recurrent networks) and Darcet et al. (2023) (for vision transformers) insert a number of special `<pause>` or "register tokens" before the model's response. We use this class of methods as a baseline in Sec. 4. Finally, Gerontopoulos et al. (2025) combines the idea of multi-token prediction and pause tokens by supervising the pause tokens with multiple future tokens. In comparison, we do not use our model as a multi-token predictor, use a different attention mask, and perform latent computation by feeding the hidden states back into the context. Finally, this idea of expanding the context is similarly explored in other works (Burtsev et al., 2020; Pfau et al., 2024).

## 3   METHODOLOGY

In autoregressive language modeling, we are given a sequence of one-hot encoded tokens $x = (x_1, \ldots x_T)$, with $x_i \in \mathbb{R}^V$, where $V$ is the vocabulary size. The aim is to learn the joint distribution

over tokens with the following factorization:

$$p(x_1, \ldots, x_T) = \prod_i p(x_{i+1}|x_{\leq i}), \tag{1}$$

where $T \in \mathbb{N}$ denotes the sequence length. We stress that this is only one of the possible factorizations for the joint distribution. The factorization in Eq. 1 translates to the so-called next token prediction objective for the sequence $x$:

$$\mathcal{L}_{\text{NTP}} = -\sum_i \log p_\theta(x_{i+1} \mid x_{\leq i}), \tag{2}$$

where $\theta$ represents the collections of all the parameters in the model. In a language model, the sequence is first embedded in a continuous space through a lookup table, giving the embedded sequence $e = (e_1, \ldots, e_T)$, where $e_i \in \mathbb{R}^D$ and $D$ is the hidden dimension. Then a (multi-layer) transformer (Vaswani et al., 2017) takes as input $e$ and produces the final hidden states $z = (z_1, \ldots, z_T)$, with the same dimensionality as $e$:

$$z_i = \text{transformer}(e_{\leq i}). \tag{3}$$

Finally, an un-embedding layer with weights $W_u \in \mathbb{R}^{D \times V}$ maps each token from the hidden dimension to the vocabulary size, and the softmax function constructs a distribution over the vocabulary:

$$p_\theta(x_{i+1}|x_{\leq i}) = \text{softmax}(W_u z_i). \tag{4}$$

Teacher-forcing is typically adopted, where the model learns $p(x_{i+1}|x_{\leq i})$ conditioned on ground-truth tokens $x_{\leq i}$, for all $i \in [T]$. During inference, the model samples autoregressively from the learned distribution, $x_{i+1} \sim p_\theta(\cdot|x_{\leq i})$, i.e. each token is produced by the model itself and appended into the context. This creates a discrepancy between inference and training, where the latter uses the ground-truth tokens as inputs instead. We illustrate inference in autoregressive models in Fig. 1.

### 3.1 LATENT LOOKAHEAD

To describe the proposed method, it is convenient to distinguish between *visible tokens* $x_i$, which correspond to the standard tokens in the training corpus, and *latent tokens* $z_{i,j}$, which represent the $j$-th latent prediction relative to a given visible token $x_i$. Visible tokens are part of the sequence observed in the data, while latent tokens are auxiliary predictions in latent space that allow the model to anticipate multiple steps ahead without committing to explicit token choices. The latent tokens are generated as follows: in the context of the notation introduced earlier, we set $z_{i,1} := z_i$, i.e. the first latent token is equal to final hidden states relative to the token $x_i$. Then, for the latent token $z_{i,j}$, let $e^z_{\leq i}$ be the expanded embedded sequence obtained by concatenating the latent tokens generated so far to the visible embeddings, i.e. $e^z_{\leq i} = (e_1, \ldots e_i, z_{i,1}, \ldots, z_{i,j-1})$. Then we set $z_{i,j}$ as:

$$z_{i,j} = \text{transformer}(e^z_{\leq i}). \tag{5}$$

The number of such autoregressive steps is set to $\tau$, the final number of latent tokens. A full group of $\tau$ latent tokens $\{z_{i,1}, \ldots, z_{i,\tau}\}$ constitutes a *latent thought*. We allow for latent lookahead at multiple positions in the sequence, as we detail later.

**Training.** In order for the model to learn how to leverage latent thoughts, we design a novel training objective to encourage lookahead. Figure 3(a) illustrates the resulting training procedure. Latent tokens are explicitly supervised against the ground-truth visible tokens $\tau$ steps ahead. Specifically, let $S$ be the set of indices of the visible tokens that are selected for the latent lookahead, with $n := |S|$ the number of positions with latent lookahead. We describe the selection strategies later. Then for any $i \in S$, the $j$-th latent token $z_{i,j}$ is trained to predict $x_{i+j}$ given the *full* latent thought:

$$\mathcal{L}_{\text{latent}} = -\sum_{i \in S} \sum_{j=1}^{\tau} \log p_\theta(z_{i,j} \to x_{i+j} \mid x_{\leq i}, \{z_{i,k}\}_{k=1}^{\tau}), \tag{6}$$

where the $z_{i,j} \to x_{i+j}$ indicates that the $x_{i+j}$ token is the label for the hidden state of $z_{i,j}$. Visible tokens $x_i$ are supervised with the standard next-token objective $\mathcal{L}_{\text{NTP}}$ in Eq. 2, but in addition we allow the visible to attend to the latent thoughts generated so far:

$$\mathcal{L}_{\text{NTP}} = -\sum_i \log p_\theta(x_{i+1} \mid x_{\leq i}, \{z_{i',j}\}_{i' \leq i, 1 \leq j \leq \tau}). \tag{7}$$

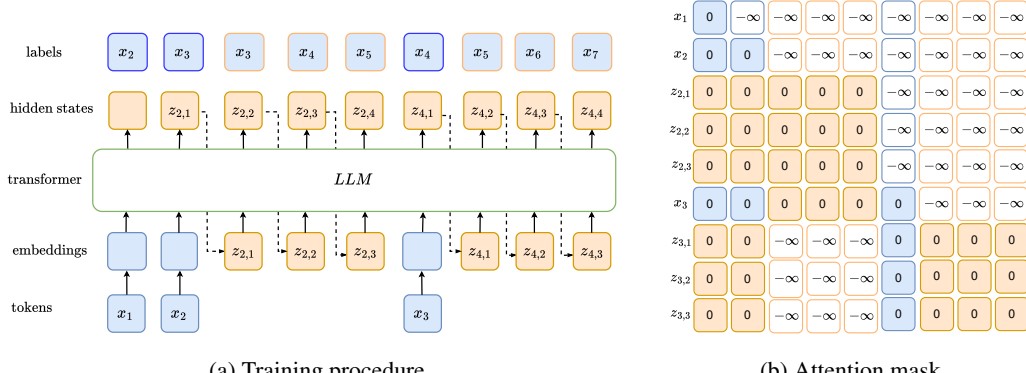

|                          |                          |
| :----------------------: | :----------------------: |
| (a) Training procedure   | (b) Attention mask       |

Figure 3: Left: the visible tokens (in blue) are supervised with standard next token prediction objective. The latent tokens are supervised explicitly with equivalent number of steps ahead, i.e. the $z_{i,j}$ latent token is supervised with the $x_{i+j}$ visible token. Right: to ensure that the model is able to refine the next token prediction based on its own assessment of the future steps, we allow for full (non-causal) masking between the latent tokens. The visible tokens follow the standard causal masking. Finally, the latent thoughts causally attend to previous visible, but not to the previous latent. This allows for parallel generation of the latent thoughts during training.

The full training objective combines the two terms:

$$\mathcal{L} = \mathcal{L}_{\text{NTP}} + \mathcal{L}_{\text{latent}}. \tag{8}$$

We remark that as we avoid sampling, our procedure is fully differentiable through the latents. Thus, backpropagation works through time (Mozer, 2013; Werbos, 2002), as in a recurrent neural network, but with an expanded context.

**Inference.** Figure 1 contrasts our latent lookahead approach with standard next-token prediction (NTP). In the standard autoregressive setting (left), the model samples a new visible token from the hidden state of the most recent token and appends it to the context: $x_{i+1} \sim p_\theta(x_{i+1}|x_{\leq i})$. In our approach (right), at selected positions the model instead enters the latent lookahead mode, generating $\tau$ latent tokens $z_{i,1}, \ldots, z_{i,\tau}$ before committing to the next visible token. The visible token is then sampled from the first latent hidden state, effectively conditioning on an internal simulation of future steps: $x_{i+1} \sim p_\theta(\cdot|x_{\leq i}, \{z_{i',j}\}_{i' \leq i, 1 \leq j \leq \tau})$. In principle, our method can be used as a multi-token predictor, decoding $\tau$ tokens after each latent thought during inference. However, our objective here is to increase the computation spent on sequence modeling by expanding the context with the latents. We perform an ablation comparing our method to multi-token prediction in the experiment section, showing that it pays off to delay the prediction, rather than predicting all the tokens at once. Per generation, our procedure adds $\mathcal{O}(n\tau)$ extra tokens to the context, with $n \ll T$.

**Attention mask.** We design a non-fully causal attention mask to handle attention between latents and visible tokens with the aim to parallelize the latent computation across the positions $\mathcal{S}$ during training. We visualize it in Figure 3(b).

*Visible-visible, latent-visible and visible-latent.* As alluded by the objective in Eq. 7, visible tokens follow standard causal masking: each $x_i$ can attend to all preceding visibles and latents. The same applies between latent and visible tokens, as well as between visible and latents. The latter is especially useful as the latents might encode important information about the model's own assessment of the future that could facilitate the decoding of future visible tokens.

*Latent-Latent (within).* Within a latent thought, we perform full (bi-directional) attention between the latent tokens. This is necessary to allow the model to refine the representation of early tokens based on its own assessment of the future ones.

*Latent-Latent (across).* A key challenge in training is to allow all latent thoughts to be generated in parallel rather than fully sequentially. Our procedure is fully sequential *within* the latent thought, requiring $\mathcal{O}(\tau)$ forward passes during training. With $n := |S|$ number of positions with latent lookahead, it would require $\mathcal{O}(\tau n)$ forward passes to compute one gradient step, which is prohibitive

Table 1: Accuracy (%) for each dataset. The number of tokens used are the same as the answer length: 19, 70, 5 for mini-Sudoku, Sudoku and ProsQA, respectively.

| Model | Mini 4×4 Sudoku | Full 9×9 Sudoku | ProsQA | Maze |
|---|---|---|---|---|
| Ours | 93.5 | 35.5 | 91.8 | 21.5 |
| Pause | 86.0 | 12.5 | 82.5 | 19.5 |
| Standard NTP | 78.0 | 12.5 | 80.5 | 18.5 |

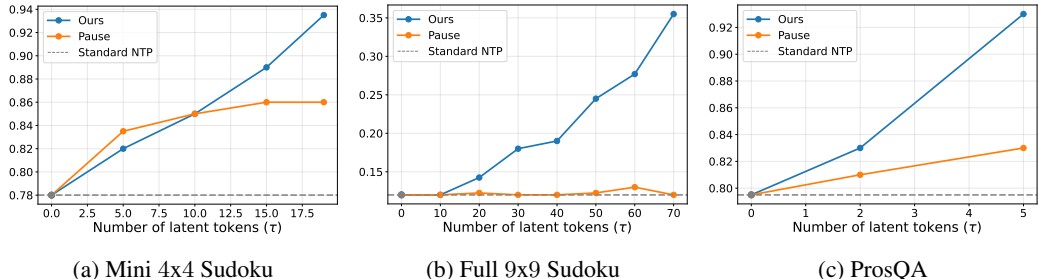

(a) Mini 4x4 Sudoku  (b) Full 9x9 Sudoku  (c) ProsQA

Figure 4: Effect of increasing the number of latent tokens.

even for relatively small models. We overcome this limitation by designing an attention mask that allows for parallel generation of the $n$ latent thoughts, reducing the complexity back to $\mathcal{O}(\tau)$ forward passes per gradient step. Specifically we achieve this by *not* allowing different latent thoughts to attend to each other. During training we generate the first token for all the latent thoughts $\{z_{i,1}\}_{i \in S}$ in parallel. Then all of them are concatenated to the corresponding visible tokens. At the next latent step, we generate $\{z_{i,2}\}_{i \in S}$, and repeat this $\tau$ times. The it ensures that the resulting computation would be the same if the procedure was performed fully sequentially. Thus, the attention mask ensures that there is no discrepancy in the computation between training and inference.

**Selecting the position of thinking tokens.** Given a fixed budget of latent thinking positions, we adopt a hybrid allocation strategy. Specifically, we always reserve one latent thought at the very beginning of the sequence (e.g., immediately after the question in reasoning tasks), ensuring that the model can initiate planning from the start. For the remaining positions, we ablate two different strategies: (1) sequentially, at each visible position. (2) uniformly at random across the sequence. In this case, at inference time, a latent thought is always inserted at the beginning, while subsequent thoughts are triggered with a fixed probability until the budget is saturated.

## 4 Experiments

The aim of our experiments is twofold. First, we aim to validate the proposed latent lookahead framework in cases where the lookahead skill is intuitive helpful, such as in solving a Sudoku. In this part, we use a smaller model trained from scratch. Second, we test whether we can incorporate the lookahead skill during the supervised fine-tuning (SFT) phase of the LLM training pipeline.

### 4.1 Latent Lookahead in Planning Tasks

We consider a two-layer GPT-2 like Transformer with hidden size 768, trained with Adam (Kingma & Ba, 2014) and constant learning of $1e-4$. We have dropout on the residual branches, embedding and attention weights to avoid overfitting. All the training details can be found in the Appendix.

**Baselines.** On top of the standard training and inference pipeline of next token prediction, we compare against *pause tokens* (Goyal et al., 2023), i.e. we insert a number of special `<pause>` tokens before the model's response. All but the last pause tokens are not supervised, and serve as a compute buffer that the model can exploit, while the last one is supervised to decode to the next token in the sequence. Notice that, in contrast to our approach, the pause tokens can be processed in parallel. We use this baseline as it processes a context of the same length. We compare against other two baselines, including the original MTP formulation of Gloeckle et al. (2024) in Appendix C.

**Datasets.** We consider the following datasets:

- **Mini-Sudoku**: We generate 2000 instances of 4x4 mini and 9x9 standard Sudoku using reasoning gym (Stojanovski et al., 2025) with between 8 and 12 empty cells. We use 1800 for training and 200 samples for the test set.

- **Full Sudoku**: Similarly, we generate 9x9 standard Sudoku instances with a number between 32 and 50 of empty cells. More difficult puzzles tend to involve a larger number of empty cells. We produce 4,000 puzzles in total and split them into training (90%) and validation (10%) sets.

- **ProsQA**(Hao et al., 2024): a datasets of directed acyclic graphs (DAGs) where given a root node and two candidate nodes, the model needs to predict to which candidate the root is connected to. The dataset also provides chain-of-thoughts in the form of sequences of nodes from root to the correct candidate. In total, we have 4-5 steps in the CoTs. Here, the model needs to plan by performing a breadth first search to scan the various branches in the graph. We use the version of the dataset from Zhu et al. (2025)

- **Maze**: we generate mazes using the library from Ivanitskiy et al. (2023). This dataset is automatically generated by constructing synthetic lattice mazes using a depth-first search (DFS)–based generator. Each maze is serialized into a tokenized string representation, where the valid solution path is enclosed between special `<PATH_START>` and `<PATH_END>` markers. From this serialization, the code extracts question–answer pairs: the full maze string serves as the question, and the path segment between the markers serves as the answer. A total of 4,000 mazes are generated on a $7 \times 7$ grid, then shuffled with a fixed seed and split into training (90%), validation (5%), and test (5%) sets.

For each dataset, we build a tokenizer with only the symbols that are needed for the tasks (i.e. integer numbers and some special characters such as delimiters).

**Results.** We use latent lookahead with $n = 1$ latent positions and a number of latent tokens $\tau$ that covers either the whole solutions (in the case of Sudoku) or the whole reasoning chain, as in the case of ProsQA. For the baseline with `<pause>` tokens, we use the same number $\tau$ of dummy tokens. The results are shown in Table 1. We observe that the model improves upon both the NTP and the `<pause>` baseline, with the largest increase on the more challenging 9x9 Sudoku, where latent lookahead improves from from 12.5% to 35%. This is also the only case where the `<pause>` tokens do not provide a benefit. These results suggest that explicitly guiding the latent tokens with the lookahead procedure provides benefit beyond the extra computation provided by the extra context.

**Scaling the Latent Computation.** Next, we analyze the role of scaling the latent computation. In our latent lookahead model, this can be done in two ways: either by increasing the number of latent tokens $\tau$ or the number of positions $n$. Given the nature of the given problems, where the very first token might require to lookahead all the way to the final part of the answer, we first analyze the former. Thus, we train our models by fixing $n = 1$ (at the first position before the answer) and all other hyperparameters and increase $\tau$. For different datasets (Sudoku and ProsQA), we use a different set of $\tau$, ranging from 2 to 70. In Fig. 4, we plot the test accuracy of the model as a function of $\tau$. Our results convincingly show that scaling $\tau$ monotonically increases the performance across the three datasets, beating both the NTP and the `<pause>` baseline, which either saturates or exhibits a worse growth rate. We report scaling $n$ in Fig. 9.

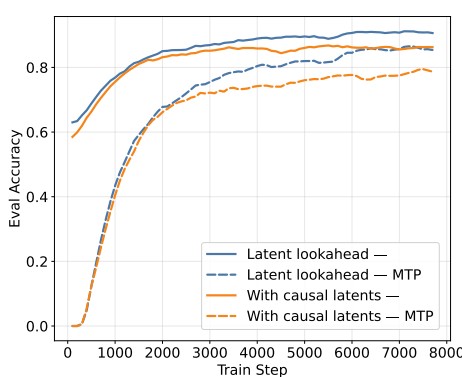

Figure 5: Eval curves ablating over the following design choices: (1) use a causal mask within the latent thoughts, and (2) use the model as a multi-token predictor (dashed).

**Ablations: On the roles of Attention Mask and Multi-Token Prediction (MTP).** We now test two key assumptions in our design. First, whether the model needs to refine the early latent tokens,

Table 2: Accuracy (%) `OLMo-2-1B` on math and logical reasoning datasets.

| Model | GSM8K | AQuA | BBH |
|---|---|---|---|
| Standard NTP | 38.0 | 24.0 | 18.2 |
| Ours ($\tau = 4$) | $39.0_{\pm0.3}$ | $23.0_{\pm0.3}$ | $18.7_{\pm0.4}$ |
| Ours ($\tau = 6$) | $37.9_{\pm0.4}$ | $26.8_{\pm0.2}$ | $18.3_{\pm0.5}$ |

Table 3: Accuracy (%) of `Qwen 2.5 0.5B` on two reasoning datasets.

| Model | Sudoku | | GSM8K | |
|---|---|---|---|---|
| | Acc. | $\tau$ | Acc. | $\tau$ |
| Standard NTP | 0.11 | – | 46.3 | – |
| Ours | 0.105 | 10 | 46.0 | 4 |
| Ours | 0.105 | 20 | 46.6 | 8 |
| Ours | 0.11 | 30 | 47.0 | 16 |

achieved with an attention mask that allows for full attention within the latent thought. To this end, we compare with the corresponding model with full causal mask across all the tokens. In principle, via backpropagation through the latents, the lookahead is still learnable by this causal model. However, this refinement cannot be made as later latents are generated. Second, we test how the proposed model performs as a multi-token predictor, where $\tau$ visible tokens are generated from the corresponding latents. This option is available as a valid inference procedure for our model, and also tests to what extent our design of latent lookahead needs an expanded context for computation to produce the solution. The results are shown in Fig. 5, where we train our model with $\tau = 19$ and plot the evaluation accuracy over training time. The experiment (1) confirms that full attention indeed induces a better latent representation that facilitates the decoding steps, and (2) shows that the model used for MTP underperforms compared to the proposed latent lookahead framework, both in the causal and full attention cases.

## 4.2 SUPERVISED FINETUNING WITH LATENT LOOKAHEAD

**Few Shot Evals on Olmo.** To assess whether latent lookahead can be incorporated into the supervised fine-tuning (SFT) phase of existing large language models, we conduct experiments starting from a pretrained `OLMo-2-1B` Base[1] model (OLMo et al., 2024). We finetune the model on the Tulu 3 SFT Mixture 0225 dataset, a diverse instruction-tuning corpus, while augmenting the training sequences with latent tokens. Specifically, we insert latent thoughts at $n = 32$ randomly chosen positions in the input sequence, and set the length of each latent thought to $\tau = 4$ latent tokens. Unlike the planning-style datasets considered in Section 4.1, here the objective is to evaluate whether latent lookahead provides benefits in more general reasoning and arithmetic tasks that are commonly used in evaluating instruction-tuned models. We evaluate the finetuned models on GSM8K (Cobbe et al., 2021), AQuA (Ling et al., 2017), and BBH (Suzgun et al., 2023), three benchmarks that probe multi-step reasoning and arithmetic problem solving. As shown in Table 2, latent lookahead yields consistent improvements over the standard next-token prediction baseline: +1% on GSM8K, +2.8% on AQuA, and +0.6% on BBH. While the gains (if any) are more modest compared to the planning tasks in Section 4.1, these results suggest that latent lookahead is a generally applicable mechanism. We further discuss this later on in the paper.

**SFT results on Qwen.** We further evaluate latent lookahead during supervised fine-tuning (SFT) of a smaller base model, Qwen2.5-0.5B (Qwen et al., 2025). Here we fine-tune directly on CoTs responses for GSM8K, which we generate with a larger Qwen2.5-7B model and filter for correct responses. For our latent lookahead, we insert latent thoughts just before the response ($n = 1$) and run a $\tau$-step latent roll-out for each. Table 3 summarizes performance on Sudoku and GSM8K as we vary the latent horizon $\tau$. We do not observe substantial improvements on Sudoku over the standard next-token-prediction (NTP) baseline. On GSM8K, latent lookahead yields gains over NTP (best +0.7% absolute), and performance improves with larger $\tau$ as well. Overall, these findings indicate that it is significantly harder to equip existing pretrained models with the latent lookahead capability. The diminishing returns we observe when applying SFT to NTP-pretrained models are consistent with prior findings in multi-token prediction and compute-increasing methods using auxiliary tokens (e.g., Pause Tokens). For example, Pause Tokens (Sec. 4.3 of their paper) yield significantly larger gains when trained from scratch than when added via SFT.

---

[1] `https://huggingface.co/allenai/OLMo-2-0425-1B`

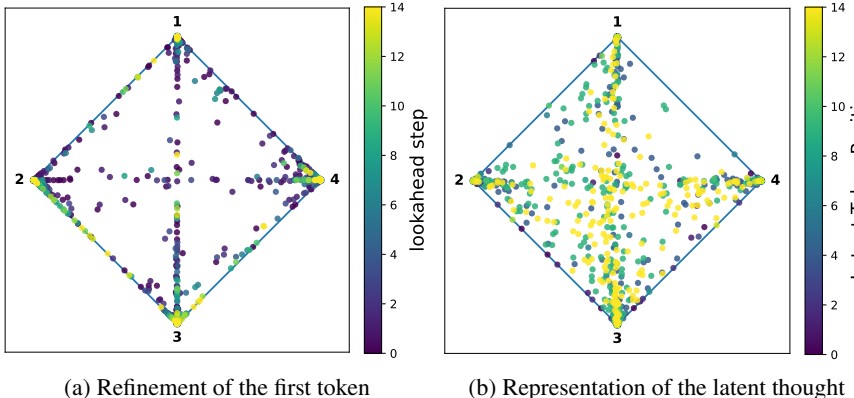

(a) Refinement of the first token  (b) Representation of the latent thought

Figure 6: Visualization of latent lookahead distributions in the probability simplex over $\{1, 2, 3, 4\}$. Each point corresponds to one latent token, projected as a convex combination of the four vertices. (a) Refinement of the first token over successive lookahead steps (color = step index). The distribution progressively sharpens towards the correct vertex, illustrating iterative refinement under the non-causal mask. (b) Representation of all latent tokens at the final step of the thought process (color = latent token position). Tokens cluster near vertices when predictions are confident, while intermediate points reflect uncertainty between candidate symbols.

### 4.3 INTERPRETING THE LATENT TOKENS

To better understand what the latent lookahead tokens represent, we visualize their distributions in a 2D simplex projection for the $4x4$ Sudoku (see Fig. 6). We map the probability mass over the four relevant symbols $\{1, 2, 3, 4\}$ to points inside a square, where each vertex corresponds to one symbol (i.e. one visible token) and the latent predictions are represented as convex combinations of these vertices. To achieve this, we process the hidden states of the latent tokens with the unembedding layer and apply the softmax to get the distribution over the visibles.

In the left panel, we track the evolution of the first latent token as additional latent steps are generated. This experiment uses the non-causal attention mask so that later latent tokens can refine earlier ones. The trajectory shows that the distribution sharpens towards one of the vertices, indicating that the latent lookahead mechanism is indeed performing iterative refinement of the prediction. In the right panel, we represent all latent tokens at the end of the thought process. In particular, we plot the final distributions of all latent tokens after the lookahead procedure has completed. Each point corresponds to one latent token in the sequence. The visualization reveals that many tokens collapse near vertices, signaling confident symbol predictions, while others remain in-between, reflecting uncertainty across multiple candidates. Due to our latent thinking, the information encoded in this superposition of states is not discarded by sampling, but kept throughout the generation process.

Overall, these visualizations support our interpretation of latent lookahead as a mechanism for iterative refinement: the model explores multiple candidate continuations, gradually sharpens its beliefs, and ultimately produces latent representations that correlate with correct outputs.

### 5 CONCLUSIONS

We introduced latent lookahead, a training strategy that enables language models to "think before talking" by unrolling hidden states into the future without committing to visible tokens. This allows models to explicitly anticipate multiple steps ahead, mitigating the limitations of standard next-token prediction. Due to page limitation, we defer further discussion and limitations of our work to App. A. Overall, our experiments demonstrate that latent lookahead improves reasoning performance on structured planning tasks (e.g., Sudoku), showing clear benefits over standard NTP and pause-token methods. Furthermore, we showed that the approach can be integrated into the supervised fine-tuning stage of large pretrained models, albeit with no significant improvement over the NTP baseline, similarly to our baselines. Future work might look into endowing LLMs with latent lookahead, overcoming some of the issues presented in this paper, thus opening new directions for reasoning-oriented training strategies.

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

## A  DISCUSSION AND LIMITATIONS

**Discussion**    Smaller gains in the SFT models suggest that the pretrained model has either learned implicitly to lookahead, making the explicit supervision of latent lookahaed less useful compared to models trained from scratch. These smaller gains are observed in standard benchmarks in all the related works that increase the computational capabilities of the model with either latent computation or with extra dummy tokens Goyal et al. (2023); Gerontopoulos et al. (2025). With this respect, our models show significant potential in models that are trained from scratch, and future work might investigate why diminishing return happen during SFT, and how this might be overcome.

**Limitations**    Our experiments are subject to the following limitations. First, due to computational constraints, all the SFT models are trained and evaluated with a single random seed. As a result, our reported numbers may be sensitive to initialization or data ordering effects. Second, our results on GSM8K with OLMo are slightly lower than those reported in OLMo et al. (2024). We hypothesize that this discrepancy may stem from differences in the fine-tuning setup: we use a smaller context length during training and evaluate with one-shot prompting rather than the eight-shot setting used in the original report. Finally, while our experiments demonstrate the effectiveness of latent lookahead across multiple tasks, they are conducted on relatively small- to mid-scale models. Future work is needed to validate whether the observed gains persist at larger model scales and under more extensive evaluation protocols.

**Future Directions.**    Latent lookahead introduces an inference-time compute–quality trade-off that depends on $(n, \tau)$ and task difficulty; learned scheduling of thoughts (e.g., via uncertainty, entropy, or value estimates) is a natural next step. Methodologically, two directions appear especially promising: (1) *Adaptive triggering and depth*, where the model decides *when* and *how far* to lookahead and (2) *Scaling laws for latent compute*, quantifying returns of $(n, \tau)$ as a function of model size and data regime. Finally, because latent lookahead makes the internal computation more legible (each latent is explicitly supervised), it may serve as a useful substrate for attribution and debugging of reasoning traces.

## B  EXPERIMENT DETAILS

**Latent Lookahead in Planning Tasks, Sec. 4.1**    We set the dropout for the residuals, the embedding and the attention to $0.3$ for ProsQA, $0.1$ for Sudokus and Maze. The rest of the architecural and optimization hyperparameter are the same across the datasets. In particular, we do not use a learning rate schedule (i.e. constant learning rate) or weight decay. The full list of hyperparameters is in Table. 4.

**SFT experiments, Sec. 4.2  Supervised Fine-Tuning (SFT) on Olmo.** We fine-tuned OLMO-2 models under a standard supervised fine-tuning setup. We used the official Tulu 3 SFT Mixture 0225 dataset, which consists of a diverse set of instruction–response pairs collected from multiple high-quality sources. All examples were wrapped using the OLMO chat template, i.e., user and assistant turns were formatted following the official convention. Since we trained with a context length of 512 tokens, we filtered out samples that exceeded this limit once tokenized, thereby avoiding excessive truncation. The details are shown in Table 5. For evaluation, we use the few-shot configurations and prompt prefixes specified in Table 7.

**SFT experiments on Qwen**    . We fine-tuned Qwen2.5-0.5B models under a standard supervised fine-tuning setup. We used the Sudoku and GSM8k datasets. We use constant learning rate and no warmup. The details of the hyperparameters and training setup are shown in Table 6.

## C  EXTRA EXPERIMENTS

**Ablating the Supervision Strategy**    To better isolate the contribution of latent lookahead, we include two additional baselines that capture the closest variants of existing approaches. (i) *MTP*: we adopt the original multi-token prediction formulation, where $\tau$ auxiliary heads are attached to the final hidden state and trained to predict the next $\tau$ ground-truth tokens. As in the canonical

| Parameter | Value |
|---|---|
| | *Model* |
| Architecture | 2-layer GPT-2 style Transformer |
| Hidden size | 768 |
| Attention heads | 8 |
| Layers | 2 |
| Vocabulary size | varies across datasets |
| Max sequence length | 512 |
| | *Latent Lookahead* |
| Latent tokens ($\tau$) | varied between 2 and 70 |
| Latent positions ($n$) | $n = 1$ before answer + varied number |
| Attention mask | causal for NTP and Pause baseline, Non-causal for latent lookahead |
| | *Training* |
| Batch size (per device) | 128 |
| Learning rate | 0.0001 |
| Warmup steps | 0 |
| Scheduler | constant |
| Optimizer | Adam (default hyperparameters) |
| Weight decay | 0.0 |
| Training steps | 10000 |
| Precision | bfloat16 |

Table 4: Main hyperparameters and settings used for the scratch experiments (Sudoku, ProsQA, Maze) with latent lookahead.

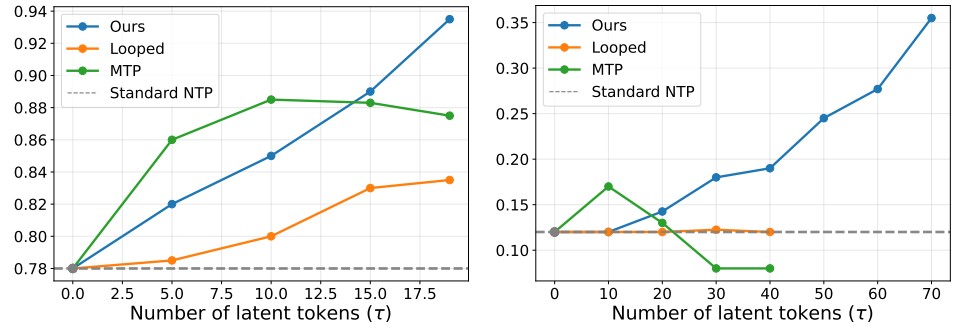

Figure 7: Ablation comparing latent lookahead with two closely related variants. Left: $4x4$ Sudoku. Standard MTP (green) improves for small $\tau$ but quickly saturates. The Looped variant (orange) performs consistently worse than our fully supervised latent lookahead (blue). In contrast, our method continues to benefit from larger $\tau$, demonstrating the value of the proposed method.

formulation, these heads are discarded at inference, so no extra computation or latent refinement is performed at generation time. This baseline captures the effect of purely adding auxiliary multi-step supervision without recursion or expanded context. (ii) *Looped*: we construct a looped-transformer-like variant of our approach by rolling the model forward for $\tau$ latent steps but removing supervision on the intermediate latent states, supervising only the last latent token. This produces an in-place iterative refinement mechanism analogous to Pause Tokens or Looped Transformers, while keeping compute identical to our full method. Overall, the results show that the improvement in performances can be attributed both to the extra computation, which alone improves over the NTP baseline, and to the supervision of the latents with $\tau$ steps ahead. However, while MTP's performance saturates, our method show a linear increase in performances with increasing $\tau$.

| Parameter | Value |
|---|---|
| Base model | `allenai/OLMo-2-0425-1B` |
| Dataset | Tulu 3 SFT (0225) |
| Hidden size | 2048 |
| Intermediate size | 8192 |
| Attention heads | 16 |
| Layers | 16 |
| Vocabulary size | 100,352 |
| Max sequence length | 512 |
| Latent tokens | 4 |
| Latent positions (max) | 32 |
| Latent embedding mode | hidden |
| Causal vis $\rightarrow$ lat | true |
| Causal lat $\rightarrow$ lat | false |
| Batch size (per device) | 2 (train), 8 (eval) |
| Gradient accumulation | 4 |
| Effective batch size | 64 tokens/device step ($2\times4\times8$ GPUs) |
| Learning rate | $1 \times 10^{-5}$ |
| Warmup steps | 1000 |
| Scheduler | linear |
| Optimizer | AdamW ($\beta_1=0.9$, $\beta_2=0.999$, $\epsilon=10^{-8}$) |
| Weight decay | 0 |
| Max steps | 40,000 |
| Precision | bfloat16 (training) |
| Gradient checkpointing | enabled |
| Hardware | $8\times$ NVIDIA A100-SXM4-40GB |

Table 5: Main hyperparameters and settings used for the `OLMo-2-0425-1B` finetuning run with latent tokens.

**Sequential vs. Random Positioning of Latent Thoughts.** We investigate the effect of different strategies for placing latent thinking tokens within the sequence. Following the setup in Figure 9, we consider two allocation schemes given a fixed budget of latent positions $\tau = 5$: Sequential: After reserving one latent token immediately after the input question, the remaining latent thoughts are inserted deterministically at subsequent visible positions. Random: After reserving the first latent token, the remaining slots are distributed uniformly at random across the sequence. At inference time, the same procedure is followed, with the initial latent token always placed after the question and the rest sampled with fixed probability $p = 0.1$ until the budget is filled. We train both strategies on the Mini Sudoku dataset, using a fixed budget of $\tau = 5$ latent tokens per latent thought. Training was conducted for 10k steps with identical optimization and model hyperparameters across conditions. The results are shown in Figure 9. Both strategies substantially outperform the standard next-token prediction (NTP) baseline, with sequential placement providing a consistent improvement over random placement when $\tau \geq 3$. This suggests that structured allocation of latent thoughts yields a more efficient use of the latent budget, leading to stronger planning and reasoning capabilities.

**Extra visualizations** In Fig. 8, we perform the same experiment as in Fig 6 but label each latent token according to whether greedy decoding matches the ground-truth target symbol, i.e. we are using the model in the MTP setting. Correct predictions are shown in green and incorrect ones in red. This highlights that the majority of latent tokens concentrate near the correct vertices, validating the utility of latent lookahead as a multi-token predictor. At the same time, the scattered red points emphasize where the refinement process fails, pointing to potential avenues for improving training signals.

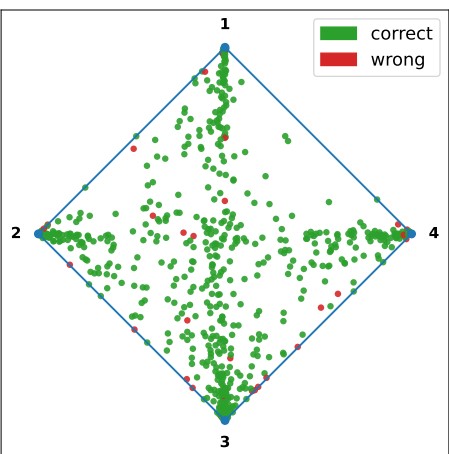

Figure 8: Visualization of Sudoku predictions. Each point corresponds to a decoded token, with green indicating a correct prediction and red a wrong one. The vertices (1–4) denote the possible output classes.

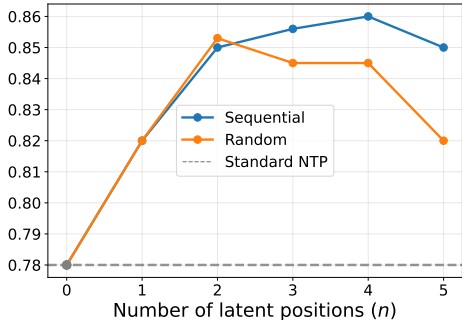

Figure 9: Comparison between sequential and random latent position strategies on mini Sudoku. Accuracy is reported as a function of the number of latent positions $n$. The sequential strategy consistently outperforms random placement, while both improve over the standard next-token prediction (dashed line).

## D  COMPUTE

All experiments are conducted on a single machine equipped with $8 \times$ NVIDIA A100-SXM4-40GB GPUs.

Table 6: Main hyperparameters and settings used for the `Qwen2.5-0.5B` finetuning run

| Parameter | Value |
|---|---|
| Base model | Qwen/Qwen2.5-0.5B |
| Dataset | Sudoku and GSM8k |
| Hidden size | 896 |
| Intermediate size | 4864 |
| Attention heads | 14 |
| Layers | 24 |
| Max sequence length | 512 |
| Latent tokens | varies |
| Latent positions | 1 |
| Batch size (per device) | 8 (train), 8 (eval) |
| Gradient accumulation | 2 |
| Effective batch size | 16 tokens/device step $\times$ 4 GPUs |
| Learning rate | $1 \times 10^{-5}$ |
| Warmup steps | 0 |
| Scheduler | constant |
| Optimizer | AdamW ($\beta_1$=0.9, $\beta_2$=0.999, $\epsilon$=$10^{-8}$) |
| Weight decay | 0 |
| Max steps | 6000 |
| Precision | float32 (training) |
| Gradient checkpointing | enabled |
| Hardware | 4 $\times$ NVIDIA A100-SXM4-40GB |

| Dataset | # shots | Shots (in order) | Prompt prefix |
|---|---|---|---|
| GSM8K | 1 | **Shot 1 — Question:** There are 15 trees in the grove. Grove workers will plant trees in the grove today. After they are done, there will be 21 trees. How many trees did the grove workers plant today? **Answer:** There are 15 trees originally. Then there were 21 trees after some more were planted. So there must have been 21 - 15 = 6. So the answer is 6. | Answer the following math question and explain step by step. |
| Aqua | 2 | **Shot 1 — Question:** Two friends plan to walk along a 43-km trail, starting at opposite ends of the trail at the same time. If Friend P's rate is 15% faster than Friend Q's, how many kilometers will Friend P have walked when they pass each other? Options: A) 21; B) 21.5; C) 22; D) 22.5; E) 23. **Answer:** If Q complete x kilometers, then P completes 1.15x kilometers. x + 1.15x = 43; 2.15x = 43; x = 43/2.15 = 20. Then P will have walked 1.15*20 = 23 km. The answer is E. | Answer the following math questions by choosing one of the options A,B,C,D,E. Explain step by step. |
| | | **Shot 2 — Question:** In the coordinate plane, points (x, 1) and (5, y) are on line k. If line k passes through the origin and has slope 1/5, then what are the values of x and y respectively? Options: A) 4 and 1; B) 1 and 5; C) 5 and 1; D) 3 and 5; E) 5 and 3. **Answer:** Line k passes through the origin and has slope 1/5, so its equation is y = (1/5)x. Thus (x,1) = (5,1) and (5,y) = (5,1) $\Rightarrow x = 5, y = 1$. The answer is C. | |
| BBH | 0 | — | Answer the following question by indicating the correct option. |

Table 7: Few-shot configurations and prompt prefixes used per dataset. Shots are taken in order from the provided pools.