# OpenReview forum: "Thinking into the Future: Latent Lookahead Training for Language Models"
_ICLR.cc/2026/Conference — Submitted to ICLR 2026_

### Official Review · Reviewer_VBXT · 2025-10-31

**Soundness:** 3
**Presentation:** 4
**Contribution:** 1
**Rating:** 2
**Confidence:** 5

**Summary:**

The paper proposes Latent Lookahead: at selected positions, the model rolls out τ steps in latent space by feeding hidden states back into the context before emitting the next visible token, and explicitly supervises each latent step against the next τ ground-truth tokens. The proposed approach is pretty similar or a combination of existing and well studied works like MTP, COCONUT and looped transformers.

**Strengths:**

- The empirical gains are clear over the standard NTP in the toy settings. However, again, most parts of the proposed approach are well studied.
- I like the proposal of parallel rollouts to make the approach more feasible.
- The writeup is pretty clear to understand.

**Weaknesses:**

- The paper proposes looping through the LLM stack to predict \tau future tokens as an auxiliary objective during training. However, this is conceptually very similar to multi-token prediction (MTP), where the model also predicts multiple future tokens. The main distinction here is that instead of a lightweight auxiliary head (as in the original MTP formulation), the authors loop through the entire transformer stack. Notably, the DeepSeek variant of MTP (https://arxiv.org/pdf/2412.19437v1) already explores an intermediate design between standard MTP and this work.


- In effect, the proposed approach can be viewed as a combination of MTP and looped transformers (https://arxiv.org/abs/2502.17416). It also bears close resemblance to COCONUT (https://arxiv.org/abs/2412.06769), which similarly performs looping in the soft-token space using intermediate supervision.


- While combining existing ideas is perfectly valid, this work does not appear to offer significant new conceptual insights or demonstrate scaling to new or large-scale setups. MTP, looped transformers, and COCONUT have all been extensively evaluated in real-world, large-model scenarios — this paper’s contribution remains mostly incremental relative to those lines of work.


- The paper critically lacks comparisons or even substantive discussion with these prior works. The brief ablation discussion at the end of Section 3.1 is inadequate. Given its close relevance, MTP should serve as a central baseline in nearly all experiments, especially since MTP has been shown to improve representation learning itself.


- The differences from MTP that are briefly mentioned in related work are insufficiently articulated and buried late in the paper. These distinctions should be clarified more prominently — ideally in the introduction — given the strong overlap in motivation and mechanism.

**Questions:**

Please check the weakness section. I will be willing to reconsider my evaluation if the authors point out a significant flaw in my understanding or highlight that I missed something. I otherwise believe the work and the proposed approach needs a significant number of ablations to tease out the differences from existing work, if at all there is any.

---

> ### Author Response · Authors · 2025-11-25
> **Response (Part I)**
>
> We thank the reviewer for the detailed assessment and for clearly articulating the concerns. We agree that latent lookahead interacts conceptually with multi-token prediction (MTP), latent-token reasoning (Coconut), and recursive computation (Looped Transformers). Below we address each point and clarify the key conceptual distinctions, including some potential misunderstandings. While we do discuss these connections in the related work, we appreciate the reviewer’s feedback and are happy to clarify the distinctions more explicitly and better position our paper.
>
> **On the relationship to MTP (standard and DeepSeek-MTP)**
>
> We agree that both our approach and MTP supervise multiple future tokens. However, the mechanisms differ in several fundamental ways. Our method’s core objective is to expand computation both along the context axis (as in Pause Tokens) and the recursion axis (as in Looped Transformers). The overlap with MTP is limited to the fact that we also supervise multiple future tokens, but the underlying mechanism is fundamentally different. However, we agree with the Reviewer that an MTP baseline would clarify some of the benefits that we observe. Thus, we add the standard formulation of MTP as a baseline (see global response). Regarding the differences with latent lookahead:
> 1. **Inference time usage**. In MTP there is no actual increase in computation during inference: the extra auxiliary heads or modules are entirely discarded (unless combined with speculative decoding like in DeepSeek). In our case, we actively use the latent tokens to generate visible tokens. Also, in MTP future visible tokens do not attend to the latent states of previous tokens, hence any speculation about future tokens is “lost” when generating new tokens. Overall, both the original MTP formulation of Gloeckle et al, 2024 and DeepSeek-MTP do not increase the computation of the model, neither in the context length axis, nor in the “recursion” axis.
> 2. **Iterative refinement**. This is a central conceptual difference: in our approach, latent tokens attend to future latent tokens (via bi-directional attention), *enabling refinement of early latent predictions using later ones*. MTP (including DeepSeek) is fully causal and cannot refine intermediate predictions. Ablations (Fig. 5) confirm that non-causal latent attention is essential for the performance gains on planning tasks such as Sudoku.
> 3. **DeepSeek-MTP shares only embeddings and output heads, while our model is fully recursive**.  DeepSeek-MTP uses new weights for each MTP module, sharing only the embedding layer and output head. Our model is fully recursive across all the network’s layers.
>
> Overall, we think that the philosophy of our method lies at the intersection between recursive computation and increasing the context. For this reason, **methods that also expand computation, such as Pause Tokens and looped-transformer variants, are more natural baselines** for our approach than MTP, whose objective is different in intent and mechanism. However, it is still interesting to analyze the sources of performance improvement over NTP, and thus having an MTP would perfectly serve this purpose. **We ablate the original MTP model** (see global response). In short, we observe that MTP provides significant benefits over NTP in the small Sudoku, but its performance improvement quickly saturates with $\tau$ and it's worse than latent lookahead. On full 9x9 Sudoku, MTP perform poorly, highlighting the difficulties with scaling to large $\tau$.

---

> ### Author Response · Authors · 2025-11-25
> **Response (Part II)**
>
> **On the relationship to Looping / Looped Transformers**
>
> We agree that both approaches involve recursive computation, but the mechanisms differ substantially.
>
> 1. Latent lookahead applies recursion selectively, which makes it more flexible and computationally scalable than looped transformers. Compared to looped transformers - where recursion is applied at every position - our method performs latent computation selectively. Furthermore, our attention masking allows us to be more efficient during training. To generate $T$ tokens during inference with $\tau$ loops at $n$ positions (where $n << T$), our method performs $O(T + n \tau)$ calls to the model, while looped transformer perform $O(T \tau)$.
> 2. Looped Transformers overwrite hidden states, allowing only for “in-place” unsupervised refinement. On the other hand, latent lookahead maintains separate latent tokens. This is a crucial step that allows us to supervise these intermediate states, incorporating a lookahead capability that standard Looped Transformers lack.
> 3. **We ablate a looped version (see global response)**: as requested, we include an ablation where we supervise only the last latent token. This may degenerate into a looped-transformer, but with an expanded context at selected positions. Thus, this method has the same computational requirements, but has a different supervision. Results show inferior performance, supporting the necessity of full latent supervision.
>
>
> **On the relationship to Coconut**
>
> Coconut replaces CoT steps with latent tokens, but it does not integrate with the next-token-prediction interface the way latent lookahead does. In particular, Coconut requires reasoning traces divided into reasoning steps due to the staged training procedure, and cannot be directly used on general SFT or pretraining data unless adopted with heuristics. Thus, it cannot be a baseline for our model, with the exception of the GSM8k and ProsQA experiments of Table 3. Regarding this, Coconut shows a significant decrease in performance on GSM8k compared to autoregressive baselines. Furthermore, during inference, Coconut relies on not generating the intermediate reasoning steps, which are entirely encapsulated into the latent tokens. On the other hand, latent lookahead expands the computation. Overall, the intended use of latent lookahead differs substantially from Coconut, and the mechanism is applicable in settings where Coconut’s staged-training constraints make it difficult to apply.
>
>
> **Summary and extra experiments**
>
> Overall, we agree with the reviewer that the comparison with MTP should be made earlier in the paper. For this reason, we moved the related work section after the introduction, rather than later in the paper.
>
> We appreciate the reviewer’s confidence in their assessment, and we hope the above clarifications demonstrate that several key assumptions, particularly about MTP equivalence, do not accurately reflect our method, which has a significant number of differences from previous work.  We believe the conceptual differences outlined above meaningfully differentiate latent lookahead from prior work and clarify the appropriate baselines for comparison.
>
> Furthermore, we hope the additional ablations (see global response) help position the paper more clearly as a hybrid method that expands computation along both the recursion (looped transformers) and context-length (Pause Tokens). We have also included a comparison with the original MTP formulation to disentangle the source of performance improvements. Overall, latent lookahead integrates ideas from these three frameworks, resulting in a method that generally scales better with the number of lookahead steps $\tau$.
>
> We sincerely thank the reviewer, and welcome further engagement.

---

### Official Review · Reviewer_T3t3 · 2025-11-01

**Soundness:** 2
**Presentation:** 3
**Contribution:** 3
**Rating:** 4
**Confidence:** 4

**Summary:**

This work proposes a latent reasoning approach that “looks ahead” a few tokens before committing to a specific token decision. This paper adopts a Coconut-style (Hao et al. 2024) approach of recurring over output latents by feeding them back into the network. A primary contribution of this paper is introducing a training procedure to make the latent recurrence more tractable compared to prior work. They report strong results on synthetic planning tasks (e.g. maze solving, Sudoku) and mixed results on real-world datasets.

**Strengths:**

This work focuses on an interesting research direction that is gaining interest recently. Namely, whether we can train models to reason directly in some latent space and obtain some benefit from that.

A big limitation of Coconut from prior work was the computational slowdown of recurrence. While this is unavoidable in some sense, this work introduces a clever training procedure to enable training with multiple latent blocks in a sequence without having to recur for every single latent token. The latent recurrences are essentially done in parallel and it simulates autoregressive generation with careful construction of the attention mask.

They adopt a reasonable selection of synthetic planning tasks to demonstrate the effectiveness of their approach. The improvements on all planning tasks are large and seem to scale well with the use of additional latent tokens. The pause token baseline is a nice inclusion.

The ablations presented in Figure 5 are nice and demonstrate that NAR refinement is beneficial and that the latent lookahead outperforms pure multi-token prediction.

The visualization of the latent thoughts (Fig 6) also provides intuition about the learned behavior of the latent thoughts.

**Weaknesses:**

All of the synthetic experiments are done with extremely shallow (2-layer) transformers. The latent lookahead essentially increases the depth of the transformer. I do wonder if the benefit extends to deeper transformers than those studies.

The experiments for SFT settings are quite mixed. No setting seems to consistently outperform SFT. When discussing the results in the paper, they take the best of two latent lookahead models (\tau=4,6). Neither one outperforms SFT consistently. As a result, I don’t think the following claim is supported by the results:
“latent lookahead yields modest but consistent improvements over autoregressive baselines“
Given the mixed SFT results, the positive results are largely constrained to synthetic tasks designed such that lookahead is useful.

It is not clear that supervising the latent tokens with the normal token labels is the obvious choice. This decision should be ablated. Is this critical for latent lookahead to work?

Although the asymptotics of the latent lookahead training are discussed, the discussion of the computational overhead during training compared to random training is not discussed. It seems, even with the efficiency improvements, that the overhead would be quite high compared to regular training, both due to the recurrence and the sequence length extension.

**Questions:**

Do the same benefits of Latent Lookahead extend to a deeper (6 layer, 12 layer, etc.) transformer?

How does the computation cost of latent lookahead compare to standard training? For the synthetic and SFT experiments?

How is training implemented? Is the entire sequence fed through for each iteration? It seems like there is significant room for training optimizations with the structured attention mask.

---

> ### Author Response · Authors · 2025-11-25
> **Response (Part I)**
>
> > All of the synthetic experiments are done with extremely shallow (2-layer) transformers. The latent lookahead essentially increases the depth of the transformer. I do wonder if the benefit extends to deeper transformers than those studies.
>
> We thank the reviewer for pointing out this interesting ablation. It is indeed interesting to compare against a standard multi-layer. We perform this ablation (**see the global reponse**) At depth 32, the baseline gets 80% on small Sudoku, whereas our method (depth=2) achieves 93.5%. Also, it is worth noticing that at $\tau=15$, our model performs 15 forward passes in a 2 layer model, which directly compares to the depth 32 experiments, while achieving 89%. Therefore the experiment suggest that increasing the depth alone does not explain the performance differences with standard MTP.
>
>
> > The experiments for SFT settings are quite mixed. No setting seems to consistently outperform SFT. When discussing the results in the paper, they take the best of two latent lookahead models (\tau=4,6). Neither one outperforms SFT consistently. As a result, I don’t think the following claim is supported by the results: “latent lookahead yields modest but consistent improvements over autoregressive baselines“ Given the mixed SFT results, the positive results are largely constrained to synthetic tasks designed such that lookahead is useful.
>
> (as in response to other reviewers). We agree with the reviewer that, across all our experiments, latent lookahead is significantly harder to make work in pretrained models during SFT than when training from scratch. Our work provides additional empirical insights into this difficulty, complementing recent efforts on recursive models for planning tasks. Notably, earlier methods such as Looped Transformers or Scaling Up Test-Time Compute with Latent Reasoning do not attempt to integrate recursive computation into pretrained models.
> Relation to existing literature:
> 1. The diminishing returns we observe when applying SFT to NTP-pretrained models are consistent with prior findings in multi-token prediction and compute-increasing methods using auxiliary tokens (e.g., Pause Tokens). For example, Pause Tokens (Sec. 4.3 of their paper) yield significantly larger gains when trained from scratch than when added via SFT.
> 2. Recent work on “Tiny Recursive Models” (TRM) [1] similarly reports that strong pretrained models - including DeepSeek R1, Claude 3.7-8k, and O3-mini-high - struggle with planning-like tasks such as Sudoku (§Table 4). Our results confirm and extend this observation: simply adding recursive computation (e.g., via latent lookahead) does not overcome these inherited limitations of pretrained NTP models.
> Taken together, our results are consistent with the broader literature on recursive computation and on extending computation through dummy or auxiliary tokens. They highlight the recurring challenge that pretrained NTP models appear strongly biased against adopting recursive or planning-like behaviors during SFT, whereas training from scratch allows these behaviors to emerge more easily.  We are in the process of adding an interpretation of our results in light of this line of work in the paper.
>
> **Speculative explanation: the pretraining prior.**
>
> We now speculate that this suggests that the next-token-prediction prior learned during pre-training is strong enough to bias the model away from performing well on these planning tasks. In the context of Sudoku, we speculate that this is due to the data-distribution shift: in the experiments in Table 3, the model is fine-tuned to produce a solution of a sudoku directly from the problem’s instance. We speculate that it is unlikely that such a data format (sudoku-instance, solution) is present in this form in the pretraining data. This makes it harder for the model to adopt recursive planning strategies.
>
>
> **We have already updated the list of contributions as follows**: *We evaluate the applicability of our framework by equipping existing base language models with latent lookahead during supervised fine-tuning (SFT). On math and logical reasoning benchmarks, latent lookahead obtains mixed results, overall matching autoregressive baselines. We report results with Olmo 2 (1B) and Qwen 2.5 (0.5B).*
> Furthermore, we have changed the discussion in the experiment section to better reflect this new perspective.

---

> > ### Author Response · Authors · 2025-11-25
> > **Response (Part II)**
> >
> > > It is not clear that supervising the latent tokens with the normal token labels is the obvious choice. This decision should be ablated. Is this critical for latent lookahead to work?
> >
> > We thank the reviewer for the insightful suggestion. The experiments with the pause tokens suggests that simply delaying the predictions with dummy (unsupervised) tokens leads to marginal benefits compared to the latent lookahead. This suggests that either (1) a form of latent computations or (2) an explicit supervision of the latent tokens leads to the observed benefits of our method. While we ablate some of the choices in Figure 5, we agree that further disentangling (1) and (2) is interesting, and would further contextualize our method in the context of existing methods, like looped transformer.
> >
> > To this end, we performed an experiment where we perform the latent computation of our (feeding the hidden states back into the context), but we do not supervise the latent tokens (similarly to Pause Tokens). **Please see the global response**. Overall, our results strongly indicate that supervising the latents with the next ground truth tokens is more effective than having the recursive steps left unsupervised.
> >
> >  > the discussion of the computational overhead during training compared to random training is not discussed.
> >
> > We would like to ask what does the reviewer mean by “random training”? Aside from this, we would like to point out that we address the computational overhead during training at multiple points. In particular, our latent lookahead procedure required $\tau$ forward passes per latent position (aka looped transformer) per batch processed. In standard autoregressive training, a single forward pass per batch is performed. In the case of thinking at multiple positions ($n$), to avoid $\tau n$ extra forward passes, we design an attention mask that allows the latents for all the positions to be generated in parallel while having consistent training and inference procedures. During inference, because we generate tokens one at a time, we require $\tau n$ extra forward passes. We are happy to further clarify any doubts, in particular if the concept of random training is elucidated.
> >
> >
> > We hope that this resolves the reviewer's concerns. We are always happy to further engage.

---

### Official Review · Reviewer_LWvo · 2025-11-01

**Soundness:** 2
**Presentation:** 3
**Contribution:** 2
**Rating:** 4
**Confidence:** 3

**Summary:**

This paper proposes training language models to do latent lookaheads where the final layers hidden state is not used to predict a token to output but instead fed back into the model as the next input token, similar to Hao et al. (2024). However, this work repeatedly passes the hidden state through the model letting the model refine multiple hidden states at a time before returning to autoregressive generation. The hidden states are supervised with the future ground truth tokens at their relative positions to the previous output token.

**Strengths:**

* Strong performance on synthetic planning benchmarks i.e. puzzles.
* Really interesting idea, building on coconut.
* Great diagrams that aid the writing in explaining their method.

**Weaknesses:**

* Their method struggles to improve performance of OLMO in real settings GSM8K, AQuA, BBH. It is unclear if this is because the method doesn't help or SFT-ing an exiting model to do latent lookahead is hard. Again their method struggles to improve Qwen via SFT. They see extremely small improvements in GSM8K and no improvement in Sudoku.
* The authors in reference to the SFT results write the following on line 396, "Overall, these findings indicate that the latent lookahead mechanism transfers to SFT models, with the largest benefits appearing on tasks that demand multi-step reasoning and planning, and incremental gains on math word problems". This statement is contradicted by the results they present in the immediately preceding paragraphs. This text needs to be revised to more accurately reflect your findings. Not everything has to work, it's better to acknowledge limitations than ignore them.
* The way latent lookaheads are supervised during training with the future tokens is surprising if not odd. Some ablation of the supervision mechanism would strengthen this work.
* While the restriction of the attention mask to prevent latent lookaheads from attending to each other is new. The mask is very clearly built upon causal and bidirectional attention. The writing is careful not to say bidirectional attention at any point, but it really should be acknowledged that within a latent lookahead block is bidirectional attention (it is weird to omit this detail - you don't want people to assume you are the first to do bidirectional attention).
* The authors claim interpretability on line 087 but do nothing to demonstrate it. The method from what I can tell is not anymore interpretable than other methods used for reasoning.
* No discussion of how to set tau and n.
* No comparison to Chain-of-Thought or Coconut.
* The interpretation of latent tokens in section 3.3 never explicitly says this is for the 4x4 sudoku puzzles, though this is what I assumed. It should be made explicit what the setting is.
* On line 199 there is an incomplete sentence "... tokens to the We perform an ..."

**Questions:**

None.

---

> ### Author Response · Authors · 2025-11-25
> **Response (Part I)**
>
> We appreciate the Reviewer acknowledgment of some of the strengths of this paper. We hope that with this response we address the main concerns.
>
> > It is unclear if this is because the method doesn't help or SFT-ing an exiting model to do latent lookahead is hard.
>
> (As in answer to LTVJ). We agree that the proposed method does not have strong performances when applied during SFT. In the case of Sudoku we observe that the model is strong during training from scratch but not while SFT-ing a pretrained model with a much larger parameter count. This suggests that the prior that the model has learned during pretraining is hard to overcome during SFT, especially in tasks (such as Sudoku), where the x,y pairs are the Sudoku instance and its solution, without language based CoT traces.
>
> > Overall, these findings indicate that the latent lookahead mechanism transfers to SFT models, with the largest benefits appearing on tasks that demand multi-step reasoning and planning, and incremental gains on math word problems". This statement is contradicted by the results they present in the immediately preceding paragraphs.
>
> We agree with the reviewer that, across all our experiments, latent lookahead is significantly harder to make work in pretrained models during SFT than when training from scratch. Our work provides additional empirical insights into this difficulty, complementing recent efforts on recursive models for planning tasks. Notably, earlier methods such as Looped Transformers or Scaling Up Test-Time Compute with Latent Reasoning do not attempt to integrate recursive computation into pretrained models.
> Relation to existing literature:
> 1. The diminishing returns we observe when applying SFT to NTP-pretrained models are consistent with prior findings in multi-token prediction and compute-increasing methods using auxiliary tokens (e.g., Pause Tokens). For example, Pause Tokens (Sec. 4.3 of their paper) yield significantly larger gains when trained from scratch than when added via SFT.
> 2. Recent work on “Tiny Recursive Models” (TRM) [1] similarly reports that strong pretrained models - including DeepSeek R1, Claude 3.7-8k, and O3-mini-high - struggle with planning-like tasks such as Sudoku (§Table 4). Our results confirm and extend this observation: simply adding recursive computation (e.g., via latent lookahead) does not overcome these inherited limitations of pretrained NTP models.
> Taken together, our results are consistent with the broader literature on recursive computation and on extending computation through dummy or auxiliary tokens. They highlight the recurring challenge that pretrained NTP models appear strongly biased against adopting recursive or planning-like behaviors during SFT, whereas training from scratch allows these behaviors to emerge more easily.  We are in the process of adding an interpretation of our results in light of this line of work in the paper.
>
> **Speculative explanation: the pretraining prior.**
>
> We now speculate that this suggests that the next-token-prediction prior learned during pre-training is strong enough to bias the model away from performing well on these planning tasks. In the context of Sudoku, we speculate that this is due to the data-distribution shift: in the experiments in Table 3, the model is fine-tuned to produce a solution of a sudoku directly from the problem’s instance. We speculate that it is unlikely that such a data format (sudoku-instance, solution) is present in this form in the pretraining data. This makes it harder for the model to adopt recursive planning strategies.
>
>
> **We have already updated the list of contributions as follows**: *We evaluate the applicability of our framework by equipping existing base language models with latent lookahead during supervised fine-tuning (SFT). On math and logical reasoning benchmarks, latent lookahead obtains mixed results, overall matching autoregressive baselines. We report results with Olmo 2 (1B) and Qwen 2.5 (0.5B).*
> Furthermore, we have changed the discussion in the experiment section to better reflect this new perspective (and will add other modifications).
>
> We sincerely thank the reviewer for the feedback on the matter, we hope that this resolves the concern.

---

> > ### Author Response · Authors · 2025-11-25
> > **Response (Part II)**
> >
> > > The way latent lookaheads are supervised during training with the future tokens is surprising if not odd.
> >
> > The latent tokens are supervised with multiple tokens ahead in the sequence, reflecting the intuitive notion of “looking ahead” before committing to the generation of one token. We would appreciate it if the Reviewer could clarify what is meant by “surprising” or “odd”, and we are happy to provide clarifications to any source of confusion.
> >
> > > The writing is careful not to say bidirectional attention at any point, but it really should be acknowledged that within a latent lookahead block is bidirectional attention
> >
> > We did not carefully avoid the term, but used “full attention” as a synonym. To make sure any confusion is avoided, we have also included “bidirectional” at multiple points in the paper and apologize if the reviewer found it hard to parse.
> >
> >
> > >  The authors claim interpretability on line 087 but do nothing to demonstrate it. The method from what I can tell is not anymore interpretable than other methods used for reasoning.
> >
> > In latent reasoning approaches, the objective is typically to internalize the reasoning process. For instance,
> > in Coconut the latent tokens compress the information of the actual CoT steps. During inference, visible CoT traces are not generated, and the model decodes the final answer from the generated latent tokens. While this is beneficial as it reduces the number of tokens generated given that the compression factor is sufficiently large, the CoT is internalized, rather than explicitly laid out. On the contrary, our latent procedure does not internalize the reasoning traces, but preserves the generation of each reasoning step.
> > As we say in the paper: "*A key design principle of our approach is interpretability: compared to existing proposed models in the realm of latent computation \cite{hao2024training, deng2024explicit}, each latent token is explicitly supervised, and the model produces explicit CoTs, ensuring that the latent computation remains transparent.* " . We hope that this clarifies the reviewer’s concern on interpretability.
> >
> >
> > > No discussion of how to set tau and n
> >
> > We ablate the effect of scaling $\tau$ in Figure 4, and the effect of scaling $n$ in Figure 8. Scaling $\tau$ provides a linear increase in performances, while when increasing $n$ the accuracy plateaus. We believe that this could be due to how our attention mask works: visible tokens can always attend to the previous lookahead buffers, while future latent thoughts cannot attend to previous ones. This might create a trade off where it becomes more beneficial to decode directly from visible tokens (as they access more information) rather than generating another latent thought. If we misunderstood the critique made the reviewer, we are happy to further engage and provide clarifications.
> >
> >
> > > No comparison to Chain-of-Thought or Coconut
> >
> > **Comparison to CoT**
> >
> > The baselines for ProsQA, and GSM8k (including Pause Tokens) are all trained with CoT traces articulating the intermediate steps. Therefore, we do compare to baselines that are trained to perform CoT. What does the reviewer mean that there is no comparison to CoT? We are happy to resolve any confusion upon further engagement
> >
> > **Comparison to Coconut**
> >
> > While our work is certainly inspired by Coconut, it significantly differs from latent lookahead. Coconut replaces CoT steps with latent tokens, but it does not integrate with the next-token-prediction interface the way latent lookahead does. In particular, Coconut requires reasoning traces divided into reasoning steps due to the staged training procedure, and cannot be directly used on general SFT or pretraining data unless adopted with heuristics. Thus, it cannot be a baseline for our model, with the exception of the GSM8k and ProsQA experiments of Table 3. On the other hand, our approach only requires a stream of tokens and does not require any curriculum. Overall, the intended use of latent lookahead significantly differs from the objectives of Coconut, which is why we have not used it as a baseline. For a deep dive into the positioning of our paper in the context of MTP, looped transformers, and Coconut, please refer to the answer to Reviewer VBXT. Our extra experiments further contextualize the source of improvements of latent lookahead compared to MTP and looped transformers (see the general response).
> >
> > We hope that this clarifies the reviewers concern. We are happy to provide more clarifications and welcome further engagement.

---

### Official Review · Reviewer_LTVJ · 2025-11-01

**Soundness:** 3
**Presentation:** 2
**Contribution:** 3
**Rating:** 4
**Confidence:** 3

**Summary:**

This paper cleverly mixes causal and non-causal language modeling in order to place more reasoning into a next token prediction by in a sense reasoning from the future. The non-causal activations are trained to match corresponding next positions in a gold standard chain of thought, so these activations gain an interpretation as corresponding to future tokens without requiring intermediate sampling. But since the LM is non-causal over these token positions, over the course of a transformer forward pass, the guesses at future tokens will use the extra computation to refine and make the next token consistent with the (predicted) future. The technique shows strong results on synthetic tasks, but not substantially improving results on GSM8K. Figure 4 hints tantalizingly at a linear improvement in prediction accuracy as a function of the number of latent lookahead tokens, though this must not hold indefinetly or we would see it in GSM8K.

**Strengths:**

* **Clever tool:** Hidden “think-then-speak” compute with dense j-step supervision; potentially useful for planning-style problems.
* **Compelling Sudoku curve:** Figure 4 shows performance improving as tau grows; if that behavior extended to long horizons on math, it would be very convincing.

**Weaknesses:**

* **Task gap:** Large delta between Sudoku (works well from scratch) and GSM8K (minimal SFT gains). Techniques that train **visible CoT** for GSM8K (e.g., RL) also work on Sudoku and don’t require training from scratch.
* **Horizon mismatch:** Likely the lookahead isn’t long enough to cover full GSM8K reasoning traces. Evaluating tau ~ 300 would be informative, but then one pass must reconcile many constraints—essentially a **100%-masked bidirectional** modeling problem, much harder than BERT’s ~15% masking.
* **Possible remedy:** Consider **recursion/iteration** within the latent block (looped/DEQ-style refinement or diffusion-style denoising) so long horizons don’t have to be solved in a single pass.
* **Presentation clarity:** Several places imply the latent lookahead is **causal**. Figures 1 and 3 use arrows that read as autoregressive; the abstract and early text say things like “recursively/iteratively feeding hidden states back … for ( \tau ) steps,” reinforcing that impression.
* **Editing issue:** The Inference paragraph on p.4 appears mid-edit (missing words), which undermines clarity about complexity and masking.

**Questions:**

# Questions

1. **Scaling:** Does this technique scale to **hundreds** of latent thinking tokens? If the trend in Figure 4 continued to tau ~ 300 for some dataset, I’d raise to a **6**.
2. **Clarity:** Can you make it explicit (in figures and text) that the **latent lookahead is non-causal** within the block while visibles remain causal? If so, I’d move to **6**; with both clarity fixes and longer-horizon/iterative results, up to **8**.

---

> ### Author Response · Authors · 2025-11-25
> **Response (Part I)**
>
> We thank the reviewer for the careful analysis and the insightful comments. We are glad that the reviewer found our approach for integrating lookahead-style computation into next-token prediction compelling, and that the results on planning tasks (e.g., Sudoku) were found strong and intuitive. We now address the main concerns, and clarify some potential misunderstanding.
>
>
> **Task gap**
>
> We agree with the reviewer that, across all our experiments, latent lookahead is significantly harder to make work in pretrained models during SFT than when training from scratch. Our work provides additional empirical insights into this difficulty, complementing recent efforts on recursive models for planning tasks. Notably, earlier methods such as Looped Transformers or Scaling Up Test-Time Compute with Latent Reasoning do not attempt to integrate recursive computation into pretrained models.
> Relation to existing literature:
> 1. The diminishing returns we observe when applying SFT to NTP-pretrained models are consistent with prior findings in multi-token prediction and compute-increasing methods using auxiliary tokens (e.g., Pause Tokens). For example, Pause Tokens (Sec. 4.3 of their paper) yield significantly larger gains when trained from scratch than when added via SFT.
> 2. Recent work on “Tiny Recursive Models” (TRM) [1] similarly reports that strong pretrained models - including DeepSeek R1, Claude 3.7-8k, and O3-mini-high - struggle with planning-like tasks such as Sudoku (§Table 4). Our results confirm and extend this observation: simply adding recursive computation (e.g., via latent lookahead) does not overcome these inherited limitations of pretrained NTP models.
> Taken together, our results are consistent with the broader literature on recursive computation and on extending computation through dummy or auxiliary tokens. They highlight the recurring challenge that pretrained NTP models appear strongly biased against adopting recursive or planning-like behaviors during SFT, whereas training from scratch allows these behaviors to emerge more easily.  We are in the process of adding an interpretation of our results in light of this line of work in the paper.
>
> **Speculative explanation: the pretraining prior.**
>
> We now speculate that this suggests that the next-token-prediction prior learned during pre-training is strong enough to bias the model away from performing well on these planning tasks. In the context of Sudoku, we speculate that this is due to the data-distribution shift: in the experiments in Table 3, the model is fine-tuned to produce a solution of a sudoku directly from the problem’s instance. We speculate that it is unlikely that such a data format (sudoku-instance, solution) is present in this form in the pretraining data. This makes it harder for the model to adopt recursive planning strategies.
>
>
> **Comparison with RL**
>
> RL-enhanced CoT approaches (e.g., ToTRL [2] or the recent Cerebras Sudoku RL system [3]) are orthogonal techniques. They can, in principle, be combined with latent lookahead; our goal is not to replace RL-based visible-CoT training but to study latent recursive computation. Thus, we view them as complementary rather than competing baselines. We believe that combining latent lookahead with RL training makes for exciting future work.
>
>
> [1] Less is More: Recursive Reasoning with Tiny Networks (https://arxiv.org/pdf/2510.04871)
>
> [2]  ToTRL: Unlock LLM Tree-of-Thoughts Reasoning Potential through Puzzles Solving (https://arxiv.org/pdf/2505.12717v1)
>
> [3] https://www.cerebras.ai/blog/from-zero-to-sudoku-hero-an-rl-adventure

---

> > ### Author Response · Authors · 2025-11-25
> > **Response (Part II)**
> >
> > **Horizon mismatch & scaling to large $\tau$**
> >
> > We are in full agreement with the reviewer that it could be that the lookahead does not span enough of the CoT trace to be sufficiently informative. We are however a bit confused about the statement that long horizons have to be resolved in a single forward pass: our method performs a forward pass per latent token (instead of a single forward pass for all the latent tokens), so conflicts can be solved iteratively and step-by-step during the latent computation.  On top of that, the model can “speculate” even more because the visible tokens are produced one at a time autoregressively while attending to this extra buffer of latent computation.
> >
> > Because of this inherently sequential latent computation, it is hard to scale our method to the requested target of 300 latent tokens. This would be closer to train a 300 times deeper model, or our 1B model would have the flops of a 300B model.  For small models we are still able to scale to 90 latent tokens in the full sudoku case, achieving a linear increase in performances with more latent computation, which suggest that the model has no difficulty in handling the bidirectional attention. In this case, we “max-out” the possibility of lookahead, as the buffer spans the whole length of the response, and thus it cannot be increased further. On GSK8K, we have to use a pretrained model, as understanding the language nuances in the question is fundamental to reply (as opposed to Sudoku).
> >
> > However, we do think that a deeper baseline can help further disentangling the sources of improvements. Yours and other reviewers’ comments suggest to try (1) deeper models, (2) recursive steps instead of lookahead (similar to looped transformers), and (3) a standard multi-token prediction baseline (Reviewer VBXT). See the global response for the results of these experiments. Overall, we observe that our model scales better with more computation than all the baseline, and depth alone cannot compensate. Although we cannot scale our model to $\tau = 300$ latents, we hope that these experiments suffice as a proof of concept.
> >
> > **Presentation Clarity**
> >
> > We appreciate the reviewer highlighting potential confusion. We will clarify the distinction explicitly.  The arrows in Figures 1-3 are intended to indicate how latent tokens are generated (solid arrows) and later concatenated to the input sequence (dashed arrows), rather than to imply that the model uses a purely autoregressive mask during latent computation. We will clarify this more explicitly in the revised version. To avoid confusion, we will emphasize two key points:
> > 1. **Generation is sequential**: the latent tokens are indeed produced one at a time by iteratively feeding the model’s hidden states back into the context. This part resembles causal computation, and we will make this explicit.
> > 2. **Refinement is non-causal**: crucially, during the latent computation we allow bi-directional attention among the latent tokens (Figure 3, right), so that earlier latent predictions can be refined using information from later latent steps. This non-causal mask is essential to our approach: it enables iterative refinement rather than a purely autoregressive rollout, and this behavior is difficult to fully capture in static figures.
> >
> > We will highlight this non-causal refinement mechanism directly in the intro to prevent confusion. In the caption of Figure 3, where we introduce the attention mask, we say “*Right: to ensure that the model is able to refine the next token prediction based on its own assessment of the future steps, we allow for full (non-causal) masking between the latent tokens*”.
> > We hope this resolves the Reviewer’s concern regarding the clarity of the relationship between sequential generation and non-causal refinement. In short, both coexist in our method. We are happy to provide further clarification or a concrete example if helpful.
> >
> > **Final remark**:
> > We appreciate the reviewer’s positive assessment of the core idea and the constructive feedback. We believe the clarified presentation, strengthened discussion of limitations, extra experiments, and the connection to recent recursive-reasoning literature will address the reviewer’s concerns.

---

### Author Response · Authors · 2025-11-25
**Extra Experiments and Ablations**

# Extra experiments:

We thank the reviewer for the their valuable feedback. In this global response, we highlight the new experiments that we believe address some of the reviewers concern and better position the contributions of this paper. We also answer to individual concerns directly.

## Comparison with Looped Transformers and MTP
To better isolate the contribution of latent lookahead, we include two additional baselines that capture the closest variants of existing approaches. (i) MTP: we adopt the original multi-token prediction formulation, where $\tau$ auxiliary heads are attached to the final hidden state and trained to predict the next $\tau$ ground-truth tokens. As in the canonical formulation, these heads are discarded at inference, so no extra computation or latent refinement is performed at generation time. This baseline captures the effect of purely adding auxiliary multi-step supervision without recursion or expanded context. (ii) Looped: we construct a looped-transformer-like variant of our approach by rolling the model forward for $\tau$ latent steps but removing supervision on the intermediate latent states, supervising only the last latent token. This produces an iterative refinement mechanism analogous to Looped Transformers, while keeping the amount of compute identical to our full method.

The results are shown in Figure 7 of the revised manuscript. Standard MTP (green) improves for small $\tau$ but quickly saturates. The Looped variant (orange) performs consistently worse than our fully supervised latent lookahead (blue). In contrast, our method continues to benefit from larger $\tau$ surpassing the MTP baseline, demonstrating the value of the proposed method. For the 9x9 Sudokus, these benefits are even more significant, where both the MTP and Looped baselines do not scale well to large $\tau$.
Overall, the results show that the performance improvement can be attributed both to the extra computation, which alone improves over the NTP baseline, and to the supervision of the latents with $\tau$ steps ahead. However, while MTP's performance saturates or entirely fails in more complicated tasks, our method shows a linear increase in performance with increasing $\tau$.

We report the numbers of Figure 7 in the table below (accuracies). The hyperparameters are identical to the rest of the experiments.

4x4 Sudoku

| $\tau$  | Looped | MTP   | Ours  |
|----|--------|--------|--------|
| 0  | 78   | 78   | 78   |
| 5  | 78.5  | 86   | 82   |
| 10 | 80   | 88.5  | 85   |
| 15 | 83   | 88.3  | 89   |
| 19 | 83.5  | 87.5  | 93.5  |


9x9 Sudoku

| $\tau$  | Looped | MTP | Ours
|----|----------|----------|----------|
| 0  | 12     | 12     | 12 |
| 10 | 12     | 17     | 12 |
| 20 | 12     | 13     | 14.5 |
| 30 | 12.2   | 8     | 18 |
| 40 | 12.2  | 8     | 19 |
| 70| -- | -- | 0.35



## Depth experiments
To test whether our gains could be explained simply by increased depth, we train baseline transformers from depth 2 to 32 on small Sudoku. As shown in the table below,, accuracy improves from 78% at depth 2 to 80% at depth 32. In contrast, our latent lookahead model, built on the same 2-layer backbone, reaches 93.5%.A compute-matched comparison leads to the same conclusion. With $\tau=15$, latent lookahead performs 15 recursive passes through a 2-layer model (effectively similar compute to a ~30-layer transformer) and achieves 89%, still far above the 80% of the actual 32-layer baseline. These results show that the improvements of latent lookahead cannot be attributed to depth alone, as these capabilities are not present in standard NTP or deeper feedforward transformers.

| Depth | Accuracy (%) |
|-------|--------------|
| 2     | 78.0         |
| 4     | 79.0         |
| 8     | 79.5         |
| 16    | 82.4         |
| 32  |  80.0        |
| Ours  | 93.5         |

---

### Meta-Review · Area_Chair_QKgY · 2025-12-12

**Summary:**

The main concerns about this paper include:

i) Lack of novelty in the latent-state idea: the core concept of introducing latent states is not new, and the paper does not sufficiently differentiate itself from prior work adopting similar formulations.

ii) Limited performance gains, especially for SFT. The improvements reported are modest, and the experiments showing non-trivial benefits are restricted to very small models, which limits the significance of the empirical findings.

iii) Missing ablation on supervising latent tokens with standard token labels. The paper does not include an ablation study investigating the effect of supervising latent tokens directly with normal token labels, which is important for understanding the contribution of the latent structure.

iv) Clarity issues and missing discussion of related work. Several aspects of the presentation lack clarity, and the literature review omits discussion of some highly relevant prior work.

In addition, after reading the paper, I identified two further questions:

i) When using $n=1$ at the beginning, is the method very close to a Masked Diffusion Model? A discussion of this connection would also help.

ii) For $n>1$, when sampling multiple latent positions at random, the strategy for selecting these positions becomes an important design choice. This aspect is not explored in the current paper and seems to require further investigation. The random choice currently used seems not perform well.

**Reviewer Concerns:**

I believe concern iv) has been adequately addressed. For the remaining concerns, the rebuttal provides some explanations, but they do not fully resolve the issues.

**Reviewer Scores:**

Reviewer LTVJ indicated that he would raise his score to 8 if the clarity issues were resolved. In my view, these issues have been fully addressed.

For the remaining reviewers, some of their concerns are partially addressed but not fully resolved. I expect that they will likely keep their current scores, though there is some possibility that one or more may increase their scores by 1.

---

### Decision · Program_Chairs · 2026-01-26

Reject